# IL-21 in Homeostasis of Resident Memory and Exhausted CD8 T Cells during Persistent Infection

**DOI:** 10.3390/ijms21186966

**Published:** 2020-09-22

**Authors:** Heather M. Ren, Aron E. Lukacher

**Affiliations:** Department of Microbiology and Immunology, The Pennsylvania State University College of Medicine, Hershey, PA 17033, USA; hschmitz@pennstatehealth.psu.edu

**Keywords:** interleukin (IL)-21, CD8 T cells, CD4 T cells, resident memory, exhaustion, persistent infection

## Abstract

CD4 T cells guide the development of CD8 T cells into memory by elaborating mitogenic and differentiation factors and by licensing professional antigen-presenting cells. CD4 T cells also act to stave off CD8 T cell dysfunction during repetitive antigen stimulation in persistent infection and cancer by mitigating generation of exhausted T cells (T_EX_). CD4 T cell help is also required for establishing and maintaining tissue-resident memory T cells (T_RM_), the nonrecirculating memory T cell subset parked in nonlymphoid tissues to provide frontline defense against reinvading pathogens. Interleukin (IL)-21 is the signature cytokine secreted by follicular helper CD4 T cells (T_FH_) to drive B cell expansion and differentiation in germinal centers to mount high-affinity, isotype class-switched antibodies. In several infection models, IL-21 has been identified as the CD4 T help needed for formation and survival of T_RM_ and T_EX_. In this review, we will explore the different memory subsets of CD8 T cells in persistent infections, the metabolic profiles associated with each, and evidence documenting the importance of CD4 T cell-derived IL-21 in regulating CD8 T_RM_ and T_EX_ development, homeostasis, and function.

## 1. Introduction

CD8 T cells are broadly demarcated into naïve, effector (T_EFF_), central memory (T_CM_), effector memory (T_EM_), resident memory (T_RM_), and exhausted (T_EX_) subsets [1,2]. These classifications are based on expression of specific surface molecules, transcription factors, enzymes, and their anatomic location (Figure 1). These distinctions may be blurred by systemic and local inflammatory conditions that allow T cells to acquire features of other subsets [3]. For example, CD8 T cells responding to persistent infections in nonlymphoid tissues often share features of both T_RM_ and T_EX_. Despite these caveats, T cell subsetting provides a useful framework to interpret T cell responses to cancer, infections, vaccines, and immunomodulatory therapies [4].

Recent literature indicates an essential role for CD4 T cells to help CD8 T cells differentiate into T_EX_ and T_RM_ during chronic systemic and CNS-localized lymphocytic choriomeningitis virus (LCMV) infections, respectively [5,6], and into a mixed T_EX_-T_RM_ state in persistent brain infections by *Toxoplasma gondii* and mouse polyomavirus (MuPyV) [7,8,9]. Recent studies from several groups have implicated interleukin (IL)-21 as the central cytokine produced by CD4 T cells that preserves the function in T_EX_ and promotes T_RM_ differentiation [10,11,12,13,14,15,16]. In light of the overlapping characteristics between T_EX_ and T_RM_ by CD8 T cells in the setting of persistent viral infections in nonlymphoid tissues, this review will focus on T_RM_ and T_EX_ during persistent infections and discuss how CD4 T cell-derived IL-21 helps CD8 T cells belonging to these subsets maximize functionality and minimize collateral tissue damage.

## 2. CD8 T Cell Memory Subsets

Memory subsets differ in migratory behavior: T_EM_ circulate through the lymphatic and blood vasculature, as well as nonlymphoid tissues, with the ability to rapidly infiltrate sites of active infection/inflammation; T_CM_ circulate through the lymph and blood; T_RM_ are permanently settled in nonlymphoid tissues; and T_EX_ can persist long-term in nonlymphoid tissues and/or lymph nodes draining tissues “contaminated” with chronic antigen (e.g., neoplasia and persistent infection) [17,18]. T_EX_ are not conventional memory T cells, but rather are comprised of transcriptionally and phenotypically distinct subsets organized in an intricate precursor–progeny relationship [19,20]. Note that there are exceptions to these working memory subset designations, such as recent evidence that T_RM_ can leave tissues and migrate to draining lymph nodes where they can transmogrify into T_CM_ and reenter the circulation [21,22,23].

As shown in Figure 1, T_EM_ express the T-box transcription factor T-bet (*Tbx21*), which has been found to regulate expression of the lymphocyte trafficking chemokine CXCR3 and the production of the lymphocyte effector cytokine IFNγ [24]. T_CM_ express Bcl2, as well as the transcription factors T cell factor 1 (TCF-1) and Eomesodermin (Eomes). Bcl2 localizes to the outer membrane of mitochondria to promote cellular survival and inhibit proapoptotic proteins such as Bax and Bak which are known to promote release of cytochrome C and reactive oxygen species. TCF-1, downstream of the canonical *Wnt* signaling pathway, binds Eomes in CD8 T cells and induces memory formation. Blimp1 (B lymphocyte-induced maturation protein 1) and its homolog Hobit (homologue of Blimp1 in T cells) are expressed by T_RM_ [25]. Blimp1, the protein encoded by *Prdm1*, and Hobit bind target sequences within the *S1pr1*, *Ccr7*, and *Tcf7* loci, which likely directly represses these genes and, in turn, prevents cell egress from the tissues [25]. In addition to TCF-1 and Eomes, T_EX_ express TCR-responsive transcription factors such as NFAT1, IRF4, BATF, and TOX that are involved in driving their dysfunctional state [26,27,28,29,30,31,32,33].

As T_EM_ migrate into tissues, lymph, and blood, they may transiently alter expression of surface receptors to allow retention in tissues. One such molecule that T_EM_ can transiently upregulate is the C-type lectin CD69. CD69 forms a complex with sphingosine phosphate-1 (S1P1), causing its internalization and destruction such that the cell no longer responds to S1P gradients (high in vasculature, low in tissues) [34]. Conversely, transcriptional downregulation of the S1P1 receptor (S1PR1) is associated with increased CD69 expression, but this may not be causally related [35]. T_CM_ notably do not express CD69 and are CD62L(L-selectin)^hi^. Like CD69 expression in certain lymphocyte subsets, expression of the CD62L lymph node homing receptor dictates migration patterns. CD62L can bind GlyCAM-1 on high endothelial venules of lymph nodes to facilitate entry into secondary lymph organs where T_CM_ are commonly found as they migrate through the lymph and blood. Like T_EM_, T_RM_ are CD62L^lo^ and express surface CD69 to promote tissue retention [36,37]. T_RM_ may also be CD103^hi^ and/or PD-1^hi^ [35,36,37,38]. The CD103(αE) subunit pairs with β7 to form an integrin heterodimer, whose ligand is E-cadherin. E-cadherin–αEβ7 binding is thought to tether T_RM_ to epithelial cells. CD103 as it relates to bT_RM_ is discussed more thoroughly below. Like CD103, PD-1 expression by T_RM_ is tissue- and pathogen-dependent [39]. CD103 is expressed by most T_RM_ in the skin, but by fewer CD8 T_RM_ in the brain [7,36,40,41,42]; PD-1 (also discussed below) is expressed by most T_RM_ in the brain but is less commonly seen on skin T_RM_ [38,39,43,44]. T_EX_ also express PD-1 as well as multiple inhibitory receptors (e.g., Lag-3, 2B4, TIM-3, and CD160) depending on the severity and duration of persistent infection [26]. The constellation of inhibitory receptors expressed by a T cell indicates the level of dysfunction, with recent work ascribing expression of inhibitory receptors to different T_EX_ stages [19,45].

The metabolic profile of each of the memory subsets differs and influences both longevity and function. T_EM_ primarily utilize aerobic glycolysis, with some reliance on oxidative phosphorylation and fatty acid uptake [34,46]. T_CM_ are biased toward oxidative phosphorylation and support fatty acid oxidation (FAO) in a tissue-dependent manner [46]. T_RM_ share a metabolic profile with T_CM_, but may uptake exogenous fatty acids from their tissue microenvironment to support FAO in a be tissue-dependent fashion [47,48]. T_EX_ have altered metabolism (i.e., dysregulated glycolysis and mitochondrial metabolism) and rely on lipolysis to fuel FAO [49]. Understanding how metabolism shapes memory T cell differentiation is an active area of investigation.

## 3. T_RM_: Regional Immunosurveillance

T_RM_ are defined by their ability to sojourn long-term in nonlymphoid tissue (e.g., skin, lung, and brain) without needing to be resupplied by circulating T_EFF_ [50]. In the antiviral response, when T_RM_ re-encounter cells expressing their cognate viral antigen, they proliferate, differentiate into effector cells with the capacity to directly kill infected cells, release antiviral molecules such as IFNγ and TNFα, and secrete chemokines to recruit other immune cells to the site of infection [51,52]. These immediate effector and sentinel alarm activities quickly act to control infections in nonlymphoid tissues [51,52], and are considered to be more effective than the memory response mounted by their circulating counterparts. There are several methods of experimentally defining a cell as T_RM_ (Table 1, [18]). Each method has its advantages, although the appropriateness of each approach may depend on the nonlymphoid tissue being studied.

Although T_RM_ have a number of generally accepted defining characteristics (i.e., maintenance in nonlymphoid tissues independent of vasculature input), they are phenotypically flexible. The tissue and pathogen context dictate expression of certain molecules by T_RM_ and may be involved in facilitating regional immunosurveillance. Some of the tissue-specific features may be attributed to whether the tissue is a barrier/mucosal site (e.g., skin, respiratory tract, female reproductive tract, or gastrointestinal tract) comprised of rapidly dividing cells to repair tissue damage, or a nonmucosal site (e.g., central nervous system), whose terminally differentiated cells necessitate a finely tuned immune response to limit tissue injury. Brain (b)T_RM_, in particular, are charged with balancing pathogen control and immunopathology. For example, LCMV-specific CD8 bT_RM_ can control LCMV intracranial (i.c.) re-infection in the absence of CD8 T_CM_ and T_EM_ [41]; in contrast, LCMV i.c. infection in naïve immunocompetent mice is lethal as a result of antiviral CD8 T cell-mediated immunopathology in the brain [56].

## 4. T_EX_: Pathogen-Specific Control

Our understanding of T_EX_ biology comes largely from studies involving experimental chronic LCMV (clone 13 and DOCILE strain) infection and mouse cancer models. An expanding body of data support the concept that T_EX_ are a distinct T cell differentiation state epigenetically programmed by repetitive TCR stimulation [57,58]. Recent work shows that T_EX_ consist of self-renewing progenitors that progress to a terminal, nonproliferative, nonfunctional state [19], with a concomitant increase in PD-1 and other inhibitory receptors [2,59,60,61]. With regard to cancer studies, CD8 T cell exhaustion is considered pathological because the tumor survives and expands in part due to the reduced effector function of exhausted CD8 T cells, as underscored by the success of checkpoint inhibitor blockade (CIB) cancer immunotherapy. Yet, the effectiveness of CIB therapy is frustratingly nonuniform among different types of cancer (e.g., melanoma and certain lung carcinomas generally respond well to CIB therapy, whereas solid tumors in the liver and colon are unlikely to respond), as well as the variable success among patients with similar types and stages of cancers. Varying degrees of antitumor T cell exhaustion at the time of CIB initiation may affect the level of response. One possibility is that CIB therapy primarily targets the progenitor T_EX_, whose frequency may vary among different types of cancer.

Recent studies have stratified the responsiveness of T_EX_ based on their expression of various molecules (e.g., PD-1 and other checkpoint inhibitory receptors, Eomes, Blimp1, TOX, and NFAT1) and the chemokine receptor CXCR5. High TCF-1 expression in CD8 T cells during LCMV Clone 13 infection has been linked to a “stem cell-like” phenotype [62]. The more practical and clinically relevant implication of these recent data is that TCF-1^hi^ PD-1^+^ CXCR5^+^ CD8 T cells, which are found primarily in the lymph nodes, respond to PD-1/PD-L1 blockade and are considered the precursors to the terminally differentiated non-PD-1/PD-L1 blockade responders [63]. Against infectious pathogens in the brain, the reduced effector function by CD8 T_EX_ may sacrifice efficiency in clearing infection, and tolerate pathogen persistence for the sake of reducing tissue damage [38,43].

## 5. CD103 in the CNS

CD103 is often used as a T_RM_ marker, albeit the fraction of T_RM_ bearing surface CD103 varies depending on the tissue examined and type of microbial pathogen studied [38]. CD103 is induced by transforming growth factor (TGF)-β signaling [36,64,65], but whether TGF-β is responsible for differences in CD103 expression by T_RM_ in different tissues remains to be determined. As stated above, the CD103 integrin is generally thought to tether T_RM_ to epithelial cells, but there is little evidence for E-cadherin expression in the CNS [42]. Furthermore, CD8 T cells isolated from brains of mice infected with *T. gondii* vary in CD103 expression, with no overt difference in their location or function between CD103^+^ and CD103^−^ populations [7,42]. CD103 expression has been linked with *T. gondii*-specific CD8 bT_RM_ that have higher TCR affinities, suggesting that its expression is related to high TCR stimulation during activation [66], but this remains correlative. A recent study of T cells in human brains also found a 40/60% split of CD103^+^/CD103^−^ CD8 T cells; there, CD103 expression correlated with upregulation of homing markers, but without a difference in localization [67]. Interestingly, in both mice and humans, there appears to be an inverse relationship between PD-1 and CD103 expression by CD8 T_RM_ [43,68]. Thus, CD103 expression is an imperfect T_RM_ marker and may not be involved in CNS localization; moreover, whether PD-1 regulates CD103 expression or possibly fosters survival of CD103^−^ CD8 T cells remains to be determined.

## 6. PD-1 as Neuroprotective

The signature inhibitory receptor for T_EX_ is PD-1. However, PD-1 does not define the cell as T_EX_ per se, because recently activated effector CD8 T cells also express PD-1 [69]. Assigning T cells to an exhausted state requires demonstrating loss of effector functionality, such as decreased cytokine production. In the MuPyV CNS infection model, PD-1 is highly expressed on CD8 bT_RM_, but not splenic CD8 T cells [38,44]. CD8 bT_RM_ generated during MCMV infection are also PD-1^hi^ [70]. Adoptive transfer studies of bT_RM_ and the finding that the *Pdcd1* locus of brain CD8 T cells, but not spleen CD8 T cells, was demethylated supports that PD-1 expression is intrinsic to bT_RM_ [38]. Nanostring gene expression analysis of the PD-L1^−/−^ brain microenvironment suggests that the PD-1:PD-L1 pathway decreases neuroinflammation during MuPyV infection [43], further indicating that PD-1 on bT_RM_ protects against CNS damage. T cells isolated from 26 post-mortem brains from patients who died of non-CNS disease showed that both PD-1 and CTLA-4 are expressed by human brain CD8 T cells, but not CD8 T cells in blood [67]. This analysis lends support to the idea that the brain environment is responsible for upregulating inhibitory receptors on CNS-infiltrating CD8 T cells. PD-1 expression appears to be limited to CD8 T cells generated against chronic viral infections as few of the CD8 T_RM_, including those in the brain, generated during acute infections express PD-1 [38,40,70,71,72]. Notably, PD-1 blockade therapy has shown benefit in progressive multifocal leukoencephalopathy, a life-threatening demyelinating brain disease caused by the JC polyomavirus [73,74]. Interestingly, PD-1 may have an impact on the metabolic profile of the cell. In gastric adenocarcinoma, for example, PD-L1 blockade resulted in an increase in fatty acid binding protein (FABP) 4/5 expression [75], suggesting that PD-1 signaling has a negative association with free fatty acid uptake and may result in the cell situated along a spectrum of T_RM_/T_EX_ rather than T_EM_/T_RM_ differentiation. Persistent antigen and/or virus-associated inflammation during chronic infections induce PD-1 on brain-infiltrating T cells. Although TCR stimulation drives PD-1 expression, maintenance of PD-1 may be antigen-independent [38]. An important area of future research will be to define the factors sustaining PD-1 expression by CD8 bT_RM_.

## 7. CD4 T Cells: The CD8 T_RM_ and T_EX_ Helpers

Multiple lines of evidence document the importance of CD4 T cell help in directing the differentiation of naïve CD8 T cells [5,6,7,9,48,76]. During priming (the accumulation of signals during the initial activation of CD8 T cells when they encounter their cognate peptide), this help is predominantly through CD4 T cell “licensing” of dendritic cells and other antigen presenting cells to express costimulatory molecules required to initiate T cell differentiation. Recent studies have revealed that CD4 T cell help also contributes to effector and memory stages of CD8 T cell differentiation [7,77].

In the female reproductive tract (FRT) during HSV-2 infection, CD4 T cells produce IFNγ to stimulate the production of the chemokines CXCL9 and CXCL10 prior to CD8 T cell entry of the tissues [78]. These chemokines bind CXCR3, which is expressed by effector CD8 T cells. CXCL9/10 gradients attract CXCR3^hi^ CD8 T cells to the FRT; it is this CD4 T cell-directed entry into the tissue that initiates T_RM_ differentiation [78]. CD4 T cell-derived IFNγ was further found to be required for CD8 T cell entry into the lung and for CD103 upregulation on the CD8 T cells during influenza infection [76], suggesting that IFNγ could both indirectly and directly initiate CD8 T_RM_ differentiation.

Coincident with reports implicating IFNγ as the CD4 T cell-derived help for CD8 T_RM_, IL-21 was described as a potential mechanism of CD4 T cell help. In 2009, a trilogy of articles in *Science* introduced the idea that IL-21 was essential for rescuing CD8 T cells from exhaustion during systemic chronic LCMV infection [10,11,12]. CD4 T cells in the brain produce IL-21 during early infection by the gliatropic JHMV coronavirus and CD8 T cells upregulate the IL-21 receptor (IL21R); notably, IL21R^−/−^ CD8 T cells have impaired granzyme B and IFNγ production [79,80]. IL-21 from CD4 T cells was also found to limit exhaustion and promote effector functionality of CD8 T cells in *T. gondii* brain infection [15]. Recently, we determined that IL-21 promotes CD8 bT_RM_ differentiation during MuPyV infection [81].

## 8. IL-21: Production and Signaling

IL-21 is primarily considered a CD4 T cell-derived cytokine, although other cell types such as NKT cells, B cells, CD8 T cells, macrophages, and dendritic cells have been reported to express low levels of *IL21* transcripts [12,82]. IL21 is the hallmark T_FH_ cytokine due to its essential role in sustaining the germinal center reaction and providing help to B cells for T-dependent antibody responses [83,84]. Yet, non-follicular CD4 T cells with T_FH_ characteristics have been described and identified as sources of IL-21 in nonlymphoid tissues. Increased numbers of circulating PD-1^hi^ CXCR5^+^ CD4 T cells are associated with elevated IL-21 in sera of patients with cystic echinococcosis [85]. CXCR5^+^ ICOS^+^ memory CD4 T cells have also been detected in the blood of multiple sclerosis patients, which is positively correlated with higher IL-21 plasma levels and Expanded Disability Status scores [86]. In the brain, CXCR5^+^ PD-1^+^ CD4 T cells have been found in a murine model of neuropsychiatric lupus, and these cells produce IL-21 and IFNγ upon ex vivo stimulation [87]. More recently, we have found that CXCR5^hi^ PD-1^hi^ CD4 T cells in the brain of MuPyV-infected mice produce IL-21 [81]. These cells have transcriptional, phenotypic, and functional overlap with T helper type 1 (T_H_1) CD4 T cells lending greater appreciation to the idea for the existence of a distinct helper CD4 T cell subset with properties shared by T_H_1 and T_FH_ cells [81]; however, lineage tracing studies are needed to determine whether these nonlymphoid CD4 T cells constitute a distinct helper subset.

TCR stimulation influences IL-21 production by CD4 T cells [82] as do particular cytokines (e.g., IFNα/β) [88]. IL-21 itself is one of the cytokines used to polarize naïve CD4 T cells into T_FH_, suggesting that IL-21 provides autocrine signaling to influence its own production [89]. IL-21 may also be produced by CD4 T cells in concert with high TCR stimulation [81,90]. Recent work by Ditoro et al. showed that IL-2 production by naïve CD4 T cells is associated with high TCR affinity and their formation into IL-21-producing T_FH_ during protein immunization and/or *Listeria monocytogenes* infection [90]. We showed that during MuPyV brain infection, IL-21-producing CD4 T cells in the CNS expressed higher affinity TCRs than the IL-21-non-producing CD4 T cells, as determined by the robust 2D micropipette adhesion assay that visualizes direct TCR:peptide-MHC ligand binding in a cell-based context (Figure 2A [81]). Notably, several genes associated with high TCR signal strength (e.g., *Icos* and *Irf4*) have been found to be upregulated in IL-21-producing CD4 T cells as compared to non-IL-21 producers [81,91]. Whether or not altering the binding affinity of the peptide-MHC class II to its TCR such that the peptide-MHC II: TCR binds more tightly actually leads to increased production of IL-21 and/or a conversion into an IL-21-producing CD4 T cell has not been formally tested.

IL21R is a common γ chain receptor (IL2R, IL4R, IL7R, IL9R, and IL15R are others). Most lymphocyte populations (NK cells, B cells, and CD8 and CD4 T cells) express IL21R. TCR stimulation has been shown to induce IL21R upregulation [92]; whether cytokines or costimulatory molecules induce IL21R expression has not been determined. IL21R activation initiates a signaling cascade through the JAK/STAT pathway, predominantly mediated by STAT1 and STAT3 [93]. STAT5 may also contribute in IL21R signal transduction, as well as phosphatidylinositol 3-kinase (PI3K) and mitogen-activated protein kinase (MAPK) [93,94]. Some of the specific target genes downstream of IL21R activation include *Gzma*, *Gzmb*, *Il10*, *Bim*, *Bcl6*, *Maf*, *Prdm1*, *Rorgt*, *Eomes*, *Socs1*, and *Socs3* [95]. Expression of genes for granzyme proteins fits with the effector-poised state of T_RM_. In addition, Eomes is associated with promoting self-renewal of memory cells [96] and is upregulated by CD8 T cells during chronic infection [97]. A recent study has shown increased expression of Eomes by an inflammatory subset of CD8 T_RM_ isolated from rectal mucosa of patients with ulcerative colitis [68].

IL21R signaling stimulates B, T, and NK cell proliferation and differentiation. In B cells, IL21R signaling can mediate proliferation and survival when combined with CD40:CD40L and BCR stimulation in the germinal center; when these other signals are absent, apoptosis may ensue [82,98,99]. In CD8 T cells and NK cells, IL21R signaling can also be proapoptotic or antiapoptotic depending on which other signaling pathways are concomitantly activated in the cell [16]. CD8 T cells stimulated with CD3 and simultaneously given IL-21 proliferate and increase IFNγ production; CD8 T cells that received IL-21 treatment alone fail to express these functions [82]. When other cytokines such as IL-2 and IL-15 were added to CD8 T cells, addition of IL-21 enhanced CD8 T cell proliferation [82]. IL-21 stimulation has also been shown to influence CD4 regulatory T cell (T_REG_) differentiation; specifically that IL-21 treatment in vitro enhanced proliferation of human CD4 CD25^−^ T cells and counteracted the suppressive activities of CD4 CD25^+^ T cells without affecting their survival [100]. Altogether, these data suggest that IL-21 tailors the immune response depending on which other stimulatory and cytokine receptors are co-engaged.

## 9. CD4 T Cell-Derived IL-21 Modulates the CD8 T Cell Response

CD4 T cell-derived IL-21 prevents CD8 T cell exhaustion during LCMV chronic infection [10,11,12], promotes CD8 T cell effector function during JHMV and *T. gondii* brain infections [15,79,80], and directs CD8 bT_RM_ differentiation and metabolism during MuPyV infection [81]. IL21R signaling as a potential regulator/promoter for oxidative metabolic pathways is also supported by work showing that IL-21 in vitro yielded CD8 T cells having an oxidative metabolic profile; specifically, IL-21 contributed to a reliance on fatty acid oxidation for fuel and a decrease in glycolysis [101]. This study further showed that IL-21 treatment decreased PD-1 expression and skewed toward a T_CM_ phenotype [101]. Interestingly, although IL-21 treatment shifted CD8 T cells to oxidative phosphorylation/FAO, IL-21-treated CD8 T cells incorporated less BODIPY dye (a measure of intracellular fatty acid levels), suggesting that these cells were more reliant on intrinsically derived fatty acids which in some contexts has been more closely associated with T_CM_ than T_RM_ [81,101].

During chronic LCMV infection, IL-21 from CD4 T cells has been shown to direct CD8 T cells to develop into a CX_3_CR1-expressing subset that is more cytotoxic, more capable of viral control, and more responsive to immunomodulatory blockade than their CX_3_CR1^−^ counterparts [13]. In *T. gondii*-infected brains, PD-1 and IL21R expression were unrelated; however, IL21R^−/−^ CD8 T cells had increased expression of other inhibitory molecules (i.e., 2B4, Lag3, and Tim-3) compared to their IL21R^+^ CD8 T cells [15], further supporting the idea that IL21R signaling in CD8 T cells offsets terminal exhaustion. The CX_3_CR1^+^ CD8 T cells were found to express transcription factors and receptors more closely aligned with T_RM_ or T_CM_ than with T_EX_ [13,102], raising the possibility that their metabolic profile may be more akin to that of T_RM_ or T_CM_. However, as shown in Figure 1 and discussed above, depending on tissue and pathogen context, these CD8 T cells may not neatly fall into a particular subset, but may instead possess mixed T_RM_/T_Ex_ or T_CM_/T_EX_ properties.

Further evidence that IL-21/IL21R signaling drives CD8 T cells toward a metabolic profile more consistent with T_RM_ comes from our recent work using MuPyV brain infection to define requirements for bT_RM_ differentiation [81]. As depicted in Figure 2B, IL21R signaling in CD8 T cells in the brains of MuPyV-infected mice drives expression of genes within the electron transport chain (ETC) [81]. Expression of these genes connotes engagement of oxidative metabolism. In line with these findings, CD8 T cells in lungs of influenza virus-infected CD4 T cell-deficient mice express fewer oxidative metabolism genes and genes in complex I of the ETC [48]. Yet to be determined is whether the CD4 T cell help for lung CD8 T_RM_ during influenza virus infection is provided by IL-21.

IL21R signaling in CD8 T cells during T_RM_ differentiation occurs with concomitant high peptide-MHC-I: TCR engagement [82]. Thus, downstream factors from both IL21R and TCR signaling cascades likely interact to determine the functional and memory potential of the cell. A number of studies point toward specific transcription factors that may integrate IL21R and TCR signaling and thereby dictate development of T_RM_. ChIP-seq genome-wide analyses have revealed that IL21R-induced STAT3 constitutively binds IRF4, a transcription factor known to increase proportional to the strength of TCR signaling [103]. Additionally, IL21R signaling induced binding of STAT3 and IRF4 to the *Prdm1* gene (encoding Blimp1) [103]. Blimp1, along with Hobit, mediates T_RM_ development in a number of tissues (Figure 1 [25]). IL21R signaling also induces BATF in CD8 T cells that may interact with TCR-induced IRF4 to maintain effector functions during chronic viral infection [14]. In addition, IL-21 also activates the PI3K-AKT-mammalian target of rapamycin (mTOR) pathway which plays an important role in T cell metabolism and differentiation [104,105]. Interestingly, mTOR contributes at least in part to regulating the expression of IRF4 in CD8 T cells, an interaction that may connect IL-21R signaling and metabolism to TCR signal strength [106,107]. Further work is needed to clarify how these transcription factors interact to guide T_RM_ differentiation.

## 10. Significance of CXCR5

CXCR5 is a chemokine receptor known for migration of B cells from the dark zone into the light zone of germinal centers via gradients of its sole ligand CXCL13, which is produced by follicular dendritic cells. CXCR5-expressing T_FH_ cells are also directed into germinal center light zones along a CXCL13 gradient. Thus, the CXCR5:CXCL13 axis colocalizes B cells with CD4 T_FH_. We found that IL-21 producing CD4 T cells in the brain of MuPyV-infected mice highly express CXCR5 (Figure 2 [81]). Brain MuPyV-specific CD8 T cells also express CXCR5 [38]. Work by the Fabry group indicated that CXCL13 was expressed by inflamed endothelial cells in the brain following an infarct and served to recruit CXCR5^+^ IL21-producing CD4 T cells to sites of inflammation [108]. Although other mechanisms may be engaged to recruit T cells to the brain (e.g., CXCL9/10: CXCR3) [7,76,78], CXCL13:CXCR5 may act to aggregate CD4 and CD8 T cells in close proximity in nonlymphoid tissues and thereby facilitate CD4 T cell crosstalk with CD8 T cells. In this connection, CXCL13^+^ cells are present in the infected cerebral ventricular lining juxtaposed with aggregates of CD4 and CD8 T cells in the periventricular zone in brains of MuPyV-infected mice (HMR, unpublished observations). In studies on CD8 T_EX_ cells during LCMV infection, CXCR5 expression has been associated with preferential expansion after PD-1 blockade [63,109] and may also signify a more functional state. The association of CXCR5 with both PD-1 expression and IL-21 signaling also raises the question whether CXCR5 expression may equate with responsiveness to PD-1 blockade via CD4 T cell-derived IL-21.

## 11. Concluding Remarks

In certain persistently infected nonlymphoid organs, T_EX_ and T_RM_ have overlapping characteristics, such as expression of common transcription factors (Eomes, Blimp1), inhibitory receptors (PD1), and chemokine receptors (CXCR5). This blurred distinction between T_EX_ and T_RM_ is particularly evident in the brain, where CD8 T cell effector activities of both subsets must be tightly tuned to control persistent infection while concomitantly limiting bystander tissue damage in an organ replete with postmitotic cells. T_EX_ and T_RM_ both exist as heterogeneous populations, with recent evidence indicating that a proliferative progenitor-to-nonproliferative terminal cell pipeline maintains each of these subsets. CD4 T cell-derived IL-21 is emerging as an important regulator of CD8 T_RM_ and T_EX_ development, homeostasis, and function. Accumulating literature suggests that IL-21 mitigates progression to end-stage, nonfunctional T_EX_ and promotes development of T_RM_ in persistently infected tissues. Recent work further supports the likelihood that IL-21 is a critical factor in sustaining this essential heterogeneity in T_EX_ and T_RM_ populations. In nonlymphoid organs where off-target tissue damage can have devastating effects (e.g., brain), IL-21 may help guide CD8 T cells toward a mixed T_RM_-T_EX_ differentiation state and improve immunotherapy outcomes for chronic infections and cancer.

## Figures and Tables

**Figure 1 ijms-21-06966-f001:**
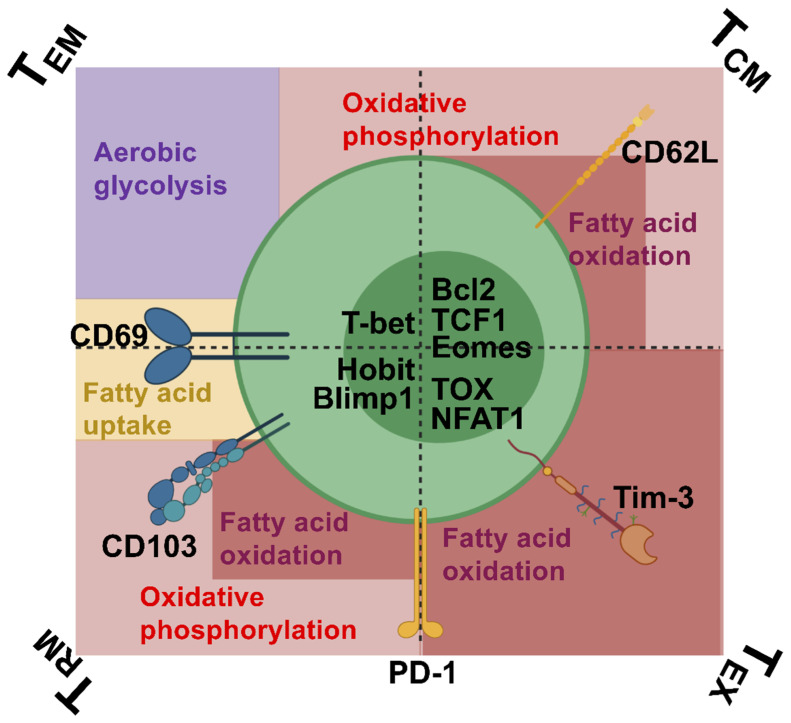
CD8 T cell memory subsets. CD8 T cells are classified as various memory subsets based on their expression of certain transcription factors, surface molecules, and metabolic profiles. As depicted here, there is fluidity and overlap in these memory profiles. Figure image created with BioRender.com.

**Figure 2 ijms-21-06966-f002:**
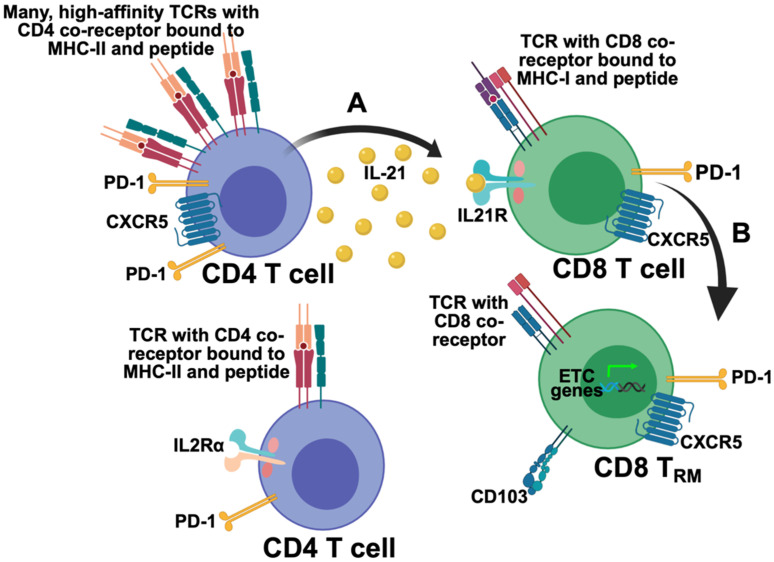
Proposed model of CD4 T cell-derived IL-21 help to CD8 T cells in the brain during PyV-infection. (**A**) CD4 T cells with a high density of high-affinity TCRs are CXCR5^hi^PD-1^hi^ and produce IL-21 that is received by the CD8 T cells in the brain. This is likely occurring around 15 dpi. (**B**) IL-21 binds to the IL21R on CD8 T cells and starts a signaling cascade that helps guide their differentiation into bT_RM_, metabolically, phenotypically, and functionally. ETC, electron transport chain. Figure image created with BioRender.com.

**Table 1 ijms-21-06966-t001:** Methods to define T_RM_.

Method	Description	Advantage(s)	Limitation(s)
Parabiosis [53]	Congenically distinct mice are joined via skin flaps that allows their vasculature to anastomose so that in ~2 weeks, blood contains equal cells of each partner	shows whether the cells are maintained independent of vascular input can be tested over many weeks can assess all organs/tissues at once	intensive procedure not appropriate for chronic infection can be difficult to interpret if the T cell population is mixed in the tissue(s) cannot assess cell egress from tissue of interest
Tissue transplant	Tissue from a congenically distinct subject is transplanted onto another	can answer whether host cells enter the transplant tissue and whether transplant cells exit	not appropriate for chronic infection inflammation induced by graft surgery could influence results only appropriate for organs that are easily transplanted (e.g., skin)
Photo-conversion [54]	Photo-convert transgenic cells (e.g., Kaede-expressing cells irreversibly change from green to red when exposed to violet light)	can assess infiltration and egress of cells from a tissue can work for chronic infection	limited by access of tissue to the violet light source protein turnover means converted fluorescent signal is lost in ~1 week
Dye-label	Inject a fluorescent dye into the tissues (e.g., CSFE) to label local cells	assesses infiltration and egress of cells from a tissue can work for chronic infection	dye diffusion away from local site can result in unintended cells getting labeled can have incomplete labeling of target cells loss of dye-labeling if cells proliferate may result in the cells of interest no longer being labelled
Intravascular antibody labeling [55]	Inject anti-CD45 (or anti-CD8, etc) i.v. 3 min prior to euthanization	reveals location of cells (vasculature vs. parenchyma) can work for chronic infection works well for highly vascularized tissue (e.g., lung)	does not reveal migration history can only determine cell location at time of euthanization
Peripheral antibody-mediated T cell depletion [44]	Inject anti-CD8 (or anti-Thy1.1, etc.) i.p. starting 8-10 dpi	can show cells are maintained independent of vasculature works well for solid organs with minimal antibody infiltration (e.g., brain, skin) can work for chronic infection can assess multiple organs/tissues at once	cannot assess cell egress from tissue(s) of interest limited use for tissues that are highly vascularized and/or mucosal if antibody can easily penetrate and deplete these cells

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
