# Peer review of "IL-21 in Homeostasis of Resident Memory and Exhausted CD8 T Cells during Persistent Infection"

_ijms, 2020, doi:10.3390/ijms21186966_

Round 1

Reviewer 1 Report

Very nice review, well organized and well written. 

Authors have a solid background in cellular immunology.

Minor point: refs 16 and 76 have to be checked.

Author Response

These references have been corrected to include the doi for the first one. For our paper in press in Science Immunology we have included the identifying information provided by the journal.

Reviewer 2 Report

This review by Ren and Lukacher describes recent advances in our understanding of IL21 and its role in exhausted and resident memory CD8+ T cells with a focus on such cells in the brain.  Overall, the topic is timely and comprehensive.  My only mild concern is that in some cases the authors could perhaps put their description of facts in better context.  For example, the paragraph spanning lines 306-314 describes a series of factors induced by IL21 but it just ends with a list of these items with no attempt to synthesize this knowledge in a succinct conclusion. I don’t have an exact request for the authors other than to perhaps ask that they reread the review and see if they can make sure that there is a conclusion to most of their paragraphs that helps underscore the main point they are trying to make. Other than that there were only a few minor issues to address (mostly typos) that are listed below.

  1. line 62 – Bcl2 is not a transcription factor. That should be corrected.
  2. line 99 – should be “support FAO in a tissue-dependent fashion” -remove “be”
  3. line 134-135 – missing “of”
  4. line 138 – “similar types of and staged cancers” – should this be “similar types and stages of cancers”??
  5. line 143 – I would list the actual molecules here, not just state “described above”.
  6. line 162 – should be “a 40/60% split” not “an 40/60% split”
  7. line 181- should be “caused by the JC” not “caused the JC”
  8. line 185 - The authors state that PD1 expression is frequently used to identify exhausted cells. It is probably true that it is used this way but the authors should mention that this is not actually a very accurate identifier as PD1 is also expressed by many recently activated effector T cells. To truly assess exhaustion one really needs to look at loss of effector function such as cytokine production. This should be clarified in the text.
  9. line 197 -should be “help also contributes” not “help is also contributes”
  10. line 249 - should be “to induce IL21R” not “to induce its IL21R”
  11. Figure 2 – please define ETC in the figure legend. My guess from later text is that this stands for electron transport chain? But it is not clear in the figure.
  12. lines 345-346 What do you mean by the phrase that the IL21R depends on concomitant pathways? It might be simpler just to state the pathways.

Author Response

Overall comment to improve putting ideas into context:  

     The paragraph spanning lines 306-314 (now lines 509-531) has been rewritten and expanded to clarify the context for introducing specific transcription factors that may integrate TCR and IL21R signaling to control development of resident-memory T cells. In addition, concluding sentences have been added to several paragraphs to summarize the information/concepts discussed.

Points 1-7, 9, and 10. We thank the reviewer for catching these typographical errors, and all have been corrected. For point 5, Tex cell molecules are now listed (line 310). 

Point 8.  The reviewer is correct that PD-1 expression doesn't define T cell exhaustion (likewise, FoxP3 alone doesn't define a CD4 T cell as a Treg). We have clarified this important point and cited a supporting recent PNAS paper from Rafi Ahmed's group that effector CD8 T cells express PD-1 (lines 345-346)

Point 11. ETC does indeed stand for electron transport chain. This abbreviation is now spelled out in the legend for Fig. 2.

Point 12. We agree. this sentence is missing details describing these "concomitant pathways" and has been deleted.

Reviewer 3 Report

The manuscript by Ren and Lukacher discuss different CD8 T cell subsets and role of IL21 in T cell memory. The review discusses T-cell metabolism and T cell exhaustion and role of Il21 in detail. The manuscript is clear and figures are presented well. Professor Lukacher is authority on T -cells immunity to polyomavirus infection. I have some minor comments.

  1. Table 1 is not clear. There are no lines to demarcate different sections. Why there is + and – sign on advantage and disadvantages. Authors should consider including references in table.

  1. In Table 1 section of Dye label, one of the disadvantages written is dye label is lost during cell proliferate. CFSE and similar dye labeling is in fact way to measure cell proliferation. Please explain.

  1. In the Peripheral antibody mediated T cell depletion authors write one disadvantage is “limited use for tissues that are highly vascularized and/or….” Is this limitation due to cost or the fact that the other methods are available? It is not clear why this is disadvantage.

  1. “During chronic LCMV Clone 13 infection, IL-21 from CD4 T cells has been shown to direct CD8 T cells to develop into” How LCMV Clone 13 infection is different from parent or other clones of LCMV? Please include details.

  1. Authors discuss metabolic control. They can consider adding a section on CD8 T cell metabolism and Il21

  1. It will be further useful if authors can provide a table with important key receptors and cytokines associated with each memory subsets in mice and humans.
  2. “Yet, despite this potential for T cell plasticity, T cell subsetting provide..” Please modify sentence.
  3. “receptors (e.g., Lag-3, 2B4. TIM-3, and CD160) depending” Please put comma after 2B4
  4. “underscored by the success checkpoint inhibitor blockade (CIB)” Please add of after success.
  5. “T cells [37,43]. bTRM generated during…” Do you mean CD8 bTRM or bTRM?

“CD4 T cell-unhelped CD8 T cells in” Please modify the sentence.

  1. Some key reference missing. PMID: 27389961, 17202333, 23064231, 25367570.

Author Response

Points 1-3.  Table 1 has been modified as suggested, with lines added to separate methods, references added, and explanations for caveats using dye label and peripheral antibody depletion approaches.

Point 4. In retrospect, we can now see how including "clone 13" as a qualifier to "chronic LCMV infection" could give the impression that this LCMV strain has some unique properties not shared with other chronic LCMV strains. Thus, "clone 13" has been removed (Line 498).

Point 5. The impact of IL-21 signaling on CD8 T cell metabolism is definitely of interest to us and others, especially in light of our new findings in the MuPyV-CNS mouse infection model (HM Ren et al. 2020. Sci Immunol).  Section 9 already discusses literature on IL-21's effects on CD8 T cell metabolic state in vitro and our work in vivo. Moreover, the Zajac Trends in Immunology review in 2016 discusses in some detail the potential effects of IL-21 on the PI3K-AKT-mTOR pathway on T cell metabolism. The last paragraph of Section 9 has been expanded to discuss the potential intersection of IL-21 with mTOR and IRF4, with two new citations added (lines 519-531).

Point 6. Given that the focus of our review is on IL-21, a table listing cytokines and receptors in both mice and humans involved in differentiation of each memory T cell subset, while useful, is a complex undertaking, particularly for TRM given differences between these molecules in different tissues.

Point 7. “Yet, despite this potential for T cell plasticity, T cell subsetting provide..” has been changed to ""Despite these caveats, T cell subsetting provides..." (line 31)

Points 8 & 9. Typographical errors fixed.

Point 10. Corrected to "CD8 bTRM" (line 348) and rewritten as "CD8 T cells in lungs of influenza virus-infected CD4 T cell-deficient mice.." (line 514)

Point 11. The indicated references have been added. Thank you.